# Morphology, Structural, Thermal, and Tensile Properties of Bamboo Microcrystalline Cellulose/Poly(Lactic Acid)/Poly(Butylene Succinate) Composites

**DOI:** 10.3390/polym13030465

**Published:** 2021-02-01

**Authors:** Masrat Rasheed, Mohammad Jawaid, Bisma Parveez, Aamir Hussain Bhat, Salman Alamery

**Affiliations:** 1Laboratory of Biocomposite Technology, Institute of Tropical Forestry and Forest Products (INTROP), Universiti Putra Malaysia, Serdang 43400, Selangor, Malaysia; masuqayyuum@gmail.com; 2Kulliyan of Engineering (KOE), Islamic International University Malaysia, Gombak 53100, Kuala Lumpur, Malaysia; mirbisma5555@gmail.com; 3Department of Applied Sciences, Higher College of Technology, University of Technology and Applied Sciences, Al-Khuwair, Muscat 133, Oman; bhataamir@gmail.com; 4Department of Biochemistry, College of Science, King Saud University, P.O. Box 22452, Riyadh 11451, Saudi Arabia; salamery@ksu.edu.sa

**Keywords:** poly(lactic acid), microcrystalline cellulose, poly(butylene succinate), scanning electron microscopy, tensile properties

## Abstract

The present study aims to develop a biodegradable polymer blend that is environmentally friendly and has comparable tensile and thermal properties with synthetic plastics. In this work, microcrystalline cellulose (MCC) extracted from bamboo-chips-reinforced poly (lactic acid) (PLA) and poly (butylene succinate) (PBS) blend composites were fabricated by melt-mixing at 180 °C and then hot pressing at 180 °C. PBS and MCC (0.5, 1, 1.5 wt%) were added to improve the brittle nature of PLA. Field emission scanning electron microscopy (FESEM), scanning electron microscopy (SEM), X-ray diffraction (XRD), Fourier transform infrared spectroscope (FTIR), thermogravimetric analysis (TGA), differential thermogravimetry (DTG), differential scanning calorimetry (DSC)), and universal testing machine were used to analyze morphology, crystallinity, physiochemical, thermal, and tensile properties, respectively. The thermal stability of the PLA-PBS blends enhanced on addition of MCC up to 1wt % due to their uniform dispersion in the polymer matrix. Tensile properties declined on addition of PBS and increased with MCC above (0.5 wt%) however except elongation at break increased on addition of PBS then decreased insignificantly on addition of MCC. Thus, PBS and MCC addition in PLA matrix decreases the brittleness, making it a potential contender that could be considered to replace plastics that are used for food packaging.

## 1. Introduction

In current years, due to the higher usage of plastics, their disposals cause environmental pollution that is of great concern. This problem can be solved by the usage of biodegradable materials which are easily disposed due to their microbial action, as a replacement to the synthetic polymers. PLA, PBS, and MCC based composites can fulfil the requirement of replacing commodity synthetic polymers. PLA and PBS are biodegradable polyesters with great degradability and mechanical properties. Since PLA behold amazing thermal, mechanical, and biodegradable properties, it holds a great potential usage in polymer-based applications [1]. In contrast, properties such as flexural properties, impact strength, gas permeability, melt viscosity required for processing, heat distortion temperature (HDT), etc. are not adequate for packaging applications [2]. Additionally, brittleness and higher cost of PLA restricts their potential for commercial usage.

Therefore, combining PLA with other appropriate biodegradable polymer with comparatively better melt processability, flexural properties, and excellent impact strength, can not only improve its properties but also reduces overall material cost. PBA is mostly preferred as it possesses the required properties. Thus, to enhance the properties and minimize the cost of production, numerous works on PLA blended with various biodegradable polymers were executed [3,4,5]. Bhatia et al. [6] fabricated a blend of PLA and PBS and observed that the blend up to 80/20 wt% (PLA-PBS) composition was partial miscibility. The brittle nature of PLA was reduced by the PBS; therefore, it can be considered as a potential material as a replacement of plastics for packaging applications. However, there was reduction of tensile strength, tensile modulus, and percentage elongation at break as PBS content increased. Liu et al. [7] repaired a blend of PLA and poly (ethylene/butylene succinate) or Binolle and observed that the addition of Bionolle resulted into crystallization of PLA and a small increment in the strain at break, however, there was reduction of ultimate tensile strength and modulus.

Also, Yokohara & Yamaguchi [8] examined the structure and various properties of PLA blended with fibrous PBS and spherical particles of PBS. PBS particle led to nucleation of PLA results into an increase in the crystallinity and it was further enhanced by the heating process. Homklin and Hongsriphan [9] reported the effect of nucleating agents such as sodium benzoate and nano-sized calcium carbonate on mechanical strength and thermal stability of PLA-PBS blends and on filling these blends with the nucleating agents, their tensile strength, energy, and elongation at break reduced. This was because of increased PLA phase crystallinity and the occurrence of stress concentration in these blends.

Microcrystalline cellulose (MCC) is a natural source with added benefits—like low cost, abundance in availability, high strength to weight ratio, renewability, and low density-makes it progressively a potential material as reinforcement for the fabrication of composites [10,11]. Although there are some challenges and further research are carried out to improve their functionalities in order to increase their applications in the materials [12]. PLA and PBS are synthetic polymers(polyesters) but MCC are the micro sized cellulose extracted mostly from plant fibers or other cellulosic waste via acid hydrolysis technique in order to eliminate their amorphous part [13,14,15]. All of the three biodegradable materials have been commercialized extensively for reduction of the number of plastics in found in the waste disposal. Even though being so beneficial they have got some limitations, low melt strength, brittleness, lower thermal stability, and lower strength that limits their application and their large-scale production [16,17,18]. However, their blends or composites may compensate their distinct shortcomings because it may lead to enhancement of properties of the developed composites.

Various researchers examined the effect of MCC on the performances of polymers composites [19,20]. Cao et al. [21] examined the effect of the content of a chain-extender on the microstructure and performance of PLA-PBS-MCC composites and observed with the incorporation of chain extender processability and strength of the composites were improved. A few researchers have previously investigated PLA-PBS-MCC composites but there is still a need to explore such composites that can in future replace plastics effectively and thus diminish the level of plastic wastes in waste disposal that otherwise lead to environmental pollution. As from the literature PLA-PBS-MCC composites are not yet studied extensively so far. In addition, these composites may also provide a new possibility of obtaining tailored and more desirable performance of biodegradable plastics, and further enhance their thermal stability and mechanical properties.

In this work, PLA-PBS-MCC composites were fabricated via the hot-pressing technique and the morphology, crystallinity, physiological, and mechanical properties and their thermal behavior were evaluated by carrying out FESEM, XRD, FTIR, tensile testing, TGA, and DSC analysis respectively. This work aims to improve our understanding and endorse the development of multi constituent materials with desirable properties and also explore the potential of MCC in packaging applications.

## 2. Experimentation

### 2.1. Materials

Pellet form PLA (7001 IngeoTM of specific gravity 1.24) utilized in this study were obtained from Nature Works LCC, MN, USA. It has melting point of 154 °C and hydrophobic in nature. PBS of density 1.26g/cm^3^ in pellet form were obtained from PTT public company limited in Bangkok, Thailand. It has a melting point of 95 °C. MCC were isolated from bamboo fiber through acid hydrolysis technique and further ultrasonicated to prevent agglomeration [22].

### 2.2. Methods

#### 2.2.1. Extraction of MCC from Bamboo Fiber

Bamboo fiber was used as a source of MCC and were extracted from it by acid hydrolysis technique. The fiber was first bleached to obtain separate lignin and hemicellulose and pre-treated bamboo pulp then was hydrolyzed with sulphuric acid of concentration 85 wt% for 30 min and finally MCC were obtained. These were further vacuum oven dried at 80 for 24 h to get dispersed and high quality MCC. Thus, MCC of higher yield of 80% and crystallinity index of 82.6% were obtained [23].

#### 2.2.2. Fabricated of MCC Reinforced PLA/PBS Composites

MCC, PBS, and PLA were initially oven-dried at a temperature of 60 °C for 24 h. Five compositions, as demonstrated in Table 1, were prepared using hot pressing technique. Initially the constituents of compositions were mixed by Brabender mixer (melt mixer) at the temperature of 180 °C for 15 min at 60 rpm. The mixture was then crushed in a crusher to get it in the form of pellets. Compressed sheets with the thickness of 0.12 mm were obtained by hot compression of these pellets in a hot press at the constant pressing pressure of 150 MPa and the temperature of 180 °C for reheating time of 4 min and then pressing time of 3 min as shown in Table 2. The compressed sheets were then oven-dried for 24 h at a constant temperature of 50 °C and prior to testing stored in desiccator.

### 2.3. Characterization and Testing

#### 2.3.1. Field Emission Scanning Electron Microscopy (FESEM)

The surface morphologies of composite films were acquired at 10–20 kV (accelerating voltage) using FESEM (JEOL JSM-7000F) from Tokyo, Japan. The films were first coated before examination to avoid electrostatic charging.

#### 2.3.2. Scanning Electron Microscopy (SEM)

The fractured surface of composite films was analyzed by SEM (Jeol, JSM 5410LV) (Hitachi Model S-3400N) from Tokyo, Japan. Specimens were tensile fracture. The samples were fixed to the stub by means of carbon tape and prior to inspection sputter-coating of gold was applied to the fractured surface to avoid electrostatic charging.

#### 2.3.3. X-ray Diffraction (XRD)

XRD analysis of composites was executed to find the crystallinity of composite films at an angular incidence of 5° to 40° using Ni-filtered Cu K-alpha radiation by SHIMADZU XRD-6000 X-ray diffractometer, from Tokyo Japan.

#### 2.3.4. Fourier Transform Infrared Spectroscope (FTIR)

The functional group analysis of each sample was examined at 4 cm^−1^ (resolution) by 32 scans using an imaging microscope (Perkin Elmer 1600 Infrared spectrometer, MA, USA) within a frequency ranging from 500 to 4000 cm^−1^

#### 2.3.5. Thermal Characterization

Thermal stability of the composite films was estimated by carrying out TGA (Perkinelmer TGA7, MA, USA) at a heat rate of 10 °C/min in range of 10–600 °C in a nitrogen atmosphere. Samples of around 10mg were cut from the films and the change in their weight ratio with respect to temperature was recorded. Thermal analysis was executed using DSC (Perkin-Elmer DSC7, MA, USA) where aluminum pan that was empty was taken as a reference. Samples of weight around 3–5 mg were placed in a pan and a temperature scan was done at a heat rate of 10 °C/min from 0–200 °C under a nitrogen atmosphere. From the thermogram attained, glass transition temperature (T_g_), crystallisation temperature (T_c_), and melting temperature (T_m_) of the composite films were obtained.

#### 2.3.6. Tensile Properties

Tensile properties were measured as per ASTM Standard Method (D638-14 (2014)) via (Lloyds LRX) Universal Testing Machine. Tensile testing was executed using samples of size (10 × 100 mm) cut from the composite films, at 50mm (initial grip separation) and 10 mm/min (crosshead speed). Ten samples of each composition were dried at 50 °C for 24 min a conventional oven prior testing. The tensile strength and modulus properties, and elongation at break were acquired from tensile testing.

## 3. Results and Discussions

### 3.1. FESEM

The surface morphology of PLA(M1), PLA-PBS(M2), and PLA-PBS-MCC (M3, M4 and M5) composites was performed as illustrated in Figure 1. PLA composite demonstrated a smoother surface, due to the poor plastic deformation [24] as evident from Figure 1a, while the homogeneously distributed PBS phase exists in all PLA-PBS blends showing the compatibility of the PLA-PBS blends [25]. The addition of PBS up to 20 wt% better compatibility can possibly be attained [1]. It can be clearly seen from Figure 1c–e, MCC was dispersed throughout PLA matrix as well as embedded within PLA matrix because of its smaller size (micro-sized) as shown in Figure 1f–h. By adding MCC and increasing the MCC content in the PLA-PBS composite, the dispersed phase of MCC becomes clearer due to lower dispersion in PLA-PBS matrix [26] and with increasing MCC content the surface topography becomes rougher, as shown in Figure 1c–e. Moreover, MCC in previous works has proved to enhances the interaction at the interfaces of the PLA and PBS blend [27,28].

### 3.2. Thermal Properties

The thermal decomposition temperature of a polymer composite can be determined by its thermal degradation behavior. TGA was executed to examine the influence of PBS and MCC’s on the thermal stability of PLA matrix composite films. The thermal stability of the composite films given by various parameters such as maximum degradation (T_max_), initial degradation (T_i_), temperature at 50% weight-loss (T50%), and final degradation (T_f_) temperatures are evaluated using TGA. Figure 2 showing TGA curves of composites with varying percentages of MCCs. All the composites exhibited degradation processes in a single step, and the parameters of their thermal stability are given in Table 3. The PLA composites exhibited relatively lower thermal decomposition temperature in comparison to PLA-PBS and PLA-PBS-MCC composites. With addition of PBS the thermal stability of PLA-PBS blend increases. The T_max_, T_i_, T50%, and T_f_, all increased on addition of up to 1 wt% MCC also similar behavior was reported by other researchers [21], then on further addition their values decrease as shown in Table 3. Even though MCC has low thermal stability when comparison with both PLA and PBS, still the char residue of MCC is expected to hinder the combustible gases to pass and diffuse into polymer matrix. Certainly, coke formed by decomposed MCC as earlier revealed to be uniformly distributed to the inside and surface of PLA-PBS-MCC composites, that efficiently impeded the discharge of decomposition products of PLA-PBS blends. Similar effect was observed for PLA-PBS-MCC blend [29] and polypropylene (PP)/MCC composites, where MCC effectively hindered the release of decomposition products of the polymers [17,30]. As evident from Table 3, PLA was found to have least residue which increased with the addition of PBS and MCC, due to their lower thermal stabilities as compared to PLA.

DTG curves determine the loss of weight and the definite temperature at which material decomposition takes place. The thermal analysis of PLA, PLA-PBS, and PLA-PBS-MCC blend with varying percentages of MCC as evident from the Figure 3a,b. The thermal degradation temperatures increased, and thereby the thermal stability of composites increased as shown in Figure 3. It was found that M1, M2, M3, M4, and M5 displayed single step degradation as evident from Figure 3. From Table 3, W_max_ decreases with the addition of PBS then on addition of MCC it further reduces insignificantly, thus exhibiting improvement of thermal stability with PBS and MCC inclusion in PLA composites.

DSC analysis characterizes the physical nature of the material. The DSC curves displayed two thermal transition temperatures: glass transition temperatures (T_g_), and the melting temperatures (T_m_) with the absence of crystallization (T_c_). The glass transition indicates a change of the degree of freedom of the molecules in the amorphous regions on increasing temperature in the polymer matrix [31]. Figure 4a,b demonstrates the DSC thermograms behavior of the composite films.

This endothermic peak values for PBS and MCC reinforced composites remain approximately contact however fluctuates to some degrees by the increase in MCC content. This endothermic peak indicates the disorder in the structure. The transition peak of PLA(M1) is obtained at 64 and on further addition of PBS and MCC in case of M2, M3, M4, and M4 the transition temperature reduces to 62 °C, 63 °C, 63.10 °C and 63.24 °C respectively as mentioned in Table 4. It reveals that the crystals in the pure PLA are more uniformly distributed as in the case of blend and also in the composites due to presence of other components during nucleation leads to crystallites varying in sizes and shapes [32]. This temperature is also relevant to the polymer’s transition temperature from glassy state to a rubbery state, i.e., with characteristics such as hard and brittle to a flexible and soft material. The T_g_ value increases with the addition of the MCC however it reduced with the addition of PBS alone in PLA composites. Also, the melting peak of PBS is obtained around 113 °C in all the composites as shown in Table 4. As per some studies PBS does not induce the crystallization [33] or improve the crystalline nature of PLA [33,34]. The impeding effect of PBS on the crystallinity of PLA was observed during isothermal crystallization around the temperature of 60 °C, indicating that PBS hinders the rate of crystallization of PLA [35]. From Figure 4a neat PLA showed two melting endothermic peaks at 151 °C and 165.14 °C. As evident from the Figure 4, the composites, M2, M3, M4, and M5 also exhibited two separate melting temperature peaks at approximately 146 °C and 153 °C as evident from Table 4 [36].

### 3.3. X-ray Diffractometry (XRD)

The crystalline structures of PLA, PLA-PBS, and PLA-PBS-MCC composite were examined by XRD in order to examine the influence of varying content of MCC on the crystallinity of the PLA matrix. The XRD patterns in 2ϴ range of 20–30° are evident from Figure 5. The pure PLA and PLA in the compositions displayed wide and indistinct diffraction patterns [37,38], indicating the existence of PLA in amorphous state mostly with small number of crystallites [38]. This may be due to lower rate of crystallization of PLA as a result faster cooling rates during processing [39,40]. Adding the PBS in the PLA resulted into small increment in the peak intensities and also in the crystalline nature of composite blends. Moreover, there was an insignificant co-crystallization among PLA and PBS [41] as also evident from DSC curve in Figure 4. Thus, PLA in the blends was not crystallized as indicated by the absence of PLA diffraction peak in all the four composites (M2, M3, M4, and M5), whereas the diffraction peak of PBS at 22.5° is present in all PLA/PBS composites that increases with increase in MCC content. This can be attributed to the diffraction peaks of both PBS and MCC (2θ = 22.6°) that overlap [42].

### 3.4. Fourier Transform Infrared Spectra (FTIR)

FTIR spectra were analyzed to study the chemical structures of composites. The FTIR spectrum of PLA(M1), PLA-PBS(M2), PLA-PBS-MCC (M3, M4, and M5) (Figure 6) displayed characteristic peak at 3750 cm^−1^ (OH stretching vibration) [43,44], and the peak intensity increases with increase in MCC content. The infrared spectra at 2993 represents the stretching vibrations of CH_3_ (asymmetric), and 2925 cm^−1^ showed the presence of stretching vibrations of CH_3_ (symmetric), respectively in case of PLA and then slowly disappears while as only 2946 cm^−1^(symmetric stretching vibration of CH_3_) peak [44,45], was observed in PLA-PBS and PLA-PBS-MCC (M3, M4, and M5) composites becoming sharper with MCC content. The wide and strong absorption peak at 1753 cm^−1^ can be associated to the presence of C=O stretching of the ester group with the addition of PBS were partially dispersed [29] and it broadens with addition of PBS as well as MCC, while as 1453 cm^-1^ represents bending vibration of CH_3_ (asymmetric) in all composites and the asymmetric C–O–C stretching mode at 1184; 1090; and 1047 cm^−1^ [39]. The vibration characteristic of the helical backbone with rocking mode of CH3 was observed at 956.2 cm^−1^ only in case of composites (M2, M3, M4, and M5). The two bands at 869 and 754 cm^−1^ presented the crystalline and amorphous phases of PLA in all compositions. The peak at 869 cm^−1^ could be represents the amorphous phase [46] and the peak at 754 cm^−1^ denotes the crystalline phase [47]. Overall, the peaks (band position) in the spectrum of PLA-PBS blend changed significantly as compared to PLA-PBS-MCC with no significant change, indicating there was no specific chemical reactions between the polymers and MCC. The curves of PLA-PBS-MCC, as shown in Figure 6, were same as that of pure PLA composite. These results show well agreement with the FESEM analysis.

### 3.5. Tensile Properties

The tensile strength and tensile modulus of PLA(M1), PLA/PBS(M2), PLA/PBS/MCC (M3, M4, and M5) composites with different MCC content are demonstrated by Figure 7a,b respectively. The tensile strength and the tensile modulus of these composite films decreased on addition of PBS and decreased initially on addition of 0.5 wt% of MCC beyond that it further increases This is due to lower tensile strength and tensile modulus of PBS than PLA. The reduced values of tensile strength may also be attributed to the weak interactions at the interfaces of PLA and PBS, resulting into the inadequate transfer of stress across each polymer phase thereby leading to quicker fracture during the tensile deformation when compared to pure PLA. Also, there was a reduction in brittleness of PLA due to decreased values of modulus from 8500 to 5400 MPa [48]. The decreased value of tensile strength at lower MCC content can be associated with the weak interactions at the interfaces of PLA-PBS matrix and MCC however it increased at higher contents of MCC due to their binding ability. The binder functionality of MCC depends upon its ability to deform plastically on application of compressive force. MCC particles form hydrogen bonds leading to strong compacts and their critical properties corresponding to their functionality as a binder include particle size, moisture content, bulk density, specific surface area, and crystallinity [49]. Similar results of decreased tensile strength were obtained by Ochi [50] in their studies. The increase in tensile strength with higher content of MCC addition in PLA-PBS matrix can be attributed to the interface action of MCC and the penetration of PLA into the network and better interaction between MCC and PLA resin as evident from FESEM images in Figure 1c–e. The increasing of the tensile modulus on addition of MCC beyond 1wt % might be due to the rigid MCC that enhances stiffness of the PLA composites by restricting the molecular movement and the distortion of PLA chains. Luz et al. [51] revealed that insertion of the fibers can increase tensile modulus due to higher modulus of fibers as compared to the modulus of thermoplastics.

The elongation at break of the composites is presented in Figure 8. It reveals that the elongation at break increases on addition of PBS and further decreases with addition of MCC insignificantly. This effect is due to the high flexibility of PBS within the PLA matrix resulting into decrease in the brittleness of the blends [52]. However, MCC inclusion reduces elongation at break but have values greater than pure PLA. Similar effect of MCC extracted from cotton waste on the PLA-PBS blend was reported by Chaiwutthinan et al. [42]. Overall, the elongation at break of the PLA composite films improves.

### 3.6. SEM Analysis of Fractured Surfaces

The tensile properties of the composites also depend upon the compatibility of PLA with PBS and MCC, they also depend on the behavior of composites when loaded. We studied the microstructures of fractured surfaces of the composites as shown in Figure 9. The microstructure indicated the dispersion and compatibility in the composites. SEM morphology of the surfaces(fractured) during tensile testing of the pure PLA as shown in Figure 9a exhibited a clear even-fractured surface and thus showed a nature of brittle material, whereas the fractured surfaces of PLA-PBS blends are rougher as evident from Figure 9b–e, suggesting more flexibility of the samples. This can be due to presence of PBS with higher toughness. The interfaces between PLA and PBS in all the blends were clear as observed in Figure 9. This reveals the immiscibility of the blend [53]. Moreover, the blends with relatively low MCC contents exhibited more interfacial debonding as a result some PBS particles are get removed completely from the composites during the fracture process as evident from Figure 9b–e. In addition, PBS particles in the composites without MCC, exhibited non-uniform distribution within PLA matrix, and thus the gaps present at the interface of PLA and PBS can enhanced water penetration by forming permeation channels that can further facilitate the hydrolytic degradation [54].

As evident from Figure 9c, the PLA-PBS-MCC cross section with 0.5 wt% MCC in the PLA-PBS matrix show the rod like structure of MCC was pulled out due to fracturing of the composite leaving a hole on the surface. These exposed MCC rods also indicate poor bonding interface thus reason of deterioration in performance of the composite. The PLA-PBS composite films with MCC exhibited a better compatibility of PLA with PBS in comparison to the composite films without MCC as evident from Figure 9b–e. However, there are MCC pulled outs that are clearly visible. On addition of more MCC (above 0.5 wt%), the MCC rods showed a well-defined and sharp edge as evident from Figure 9d,e. Also, PLA-PBS-MCC composites with MCC content more than 0.5 wt% the cross-sectional surface consists of least number of voids in comparison to composites with lower MCC content. Moreover, observed relatively compact and smooth cross-sectional surface for M4 and M5 as evident from Figure 9d,e. Therefore, there was increase in binding force of the composites on increasing concentration of MCC indicating that MCC modified the surface successfully resulting into enhanced interfacial tension [45]. On addition of 1.5 wt% of MCC, an additional fractured rod-shaped MCC were observed (Figure 9e); indicating the improvement in the cohesive forces at the interface of MCC and the PLA matrix; thus the matrix interface became more miscible as a result increases the compatibility of PLA with PBS. Thus, the incorporation of MCC improvises the interfacial complications between PLA, PBS, and MCC thereby improving mechanical properties of the composites.

## 4. Conclusions

In this research work, PLA, PLA-PBS, and PLA-PBS-MCC (0.5, 1, and 1.5 wt%) composites, where MCC extracted from bamboo fiber, were successfully prepared by hot pressing technique. Although PBS revealed insignificant impact on PLA’s crystallinity and mechanical properties however it changed PLA from brittle to flexible material that was further enhanced by the addition of MCC. In addition, MCC improved the interfacial bonding between PLA and PBS in the composites. Furthermore, MCC enhanced the crystallinity, thermal stability as well as the tensile strength of the PLA-PBS blends. The tensile strength and tensile modulus of PLA-PBS and PLA-PBS-MCC were lower at low MCC content when compared with the pure PLA composites however the elongation at break of PLA-PBS and PLA-PBS-MCC was found to be higher than that of PLA. The SEM analysis of surfaces (fractured) reveal the better bonding of MCCs in the composites containing MCC and thus exhibit better mechanical properties. Thus, the PBS and MCC-reinforced PLA matrix composites can be regarded as a potential material for packaging applications.

## Figures and Tables

**Figure 1 polymers-13-00465-f001:**
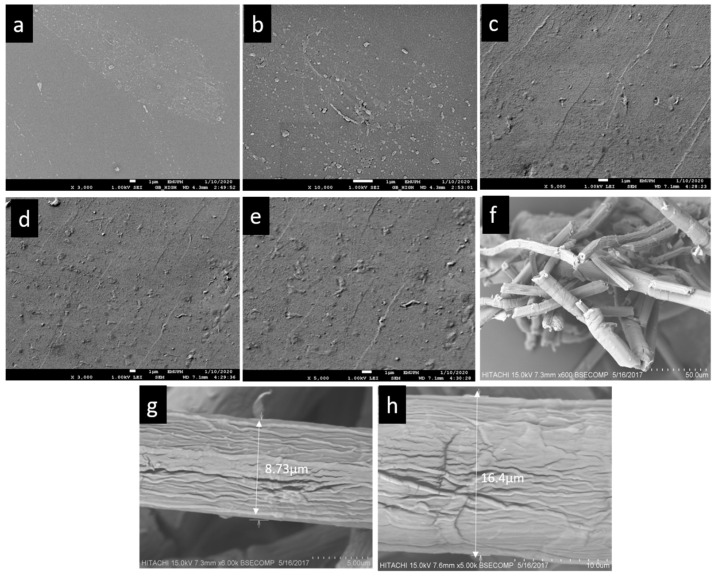
FESEM images of composite films (**a**) M1, (**b**) M2, (**c**) M3, (**d**) M4, (**e**) M5, (**f**,**g**,**h**) SEM images of MCC.

**Figure 2 polymers-13-00465-f002:**
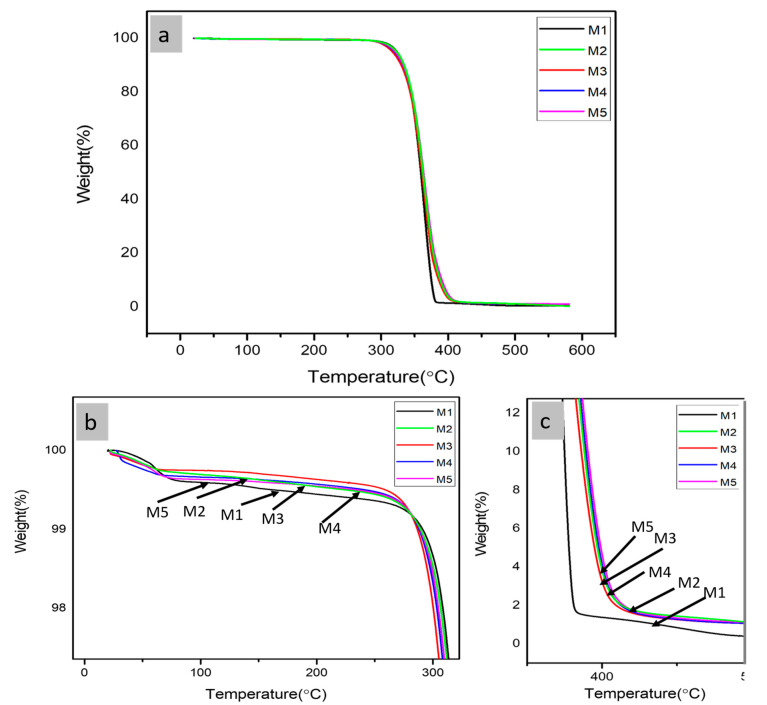
(**a**) TGA curves, (**b**) zoomed initial decomposition curves, (**c**) zoomed final decomposition curves of PLA-PBS-MCC composite films.

**Figure 3 polymers-13-00465-f003:**
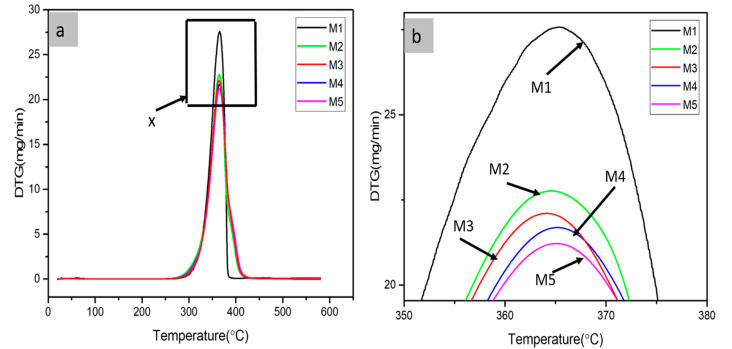
DTG of (**a**) PLA, PLA-PBS, PLA-PBS-MCC composite films, (**b**) zoomed x part.

**Figure 4 polymers-13-00465-f004:**
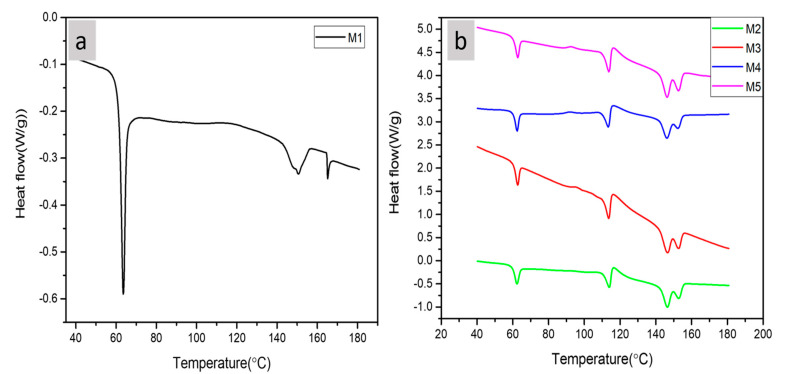
DSC of (**a**) PLA and (**b**) PLA-PBS, PLA-PBS-MCC composite films.

**Figure 5 polymers-13-00465-f005:**
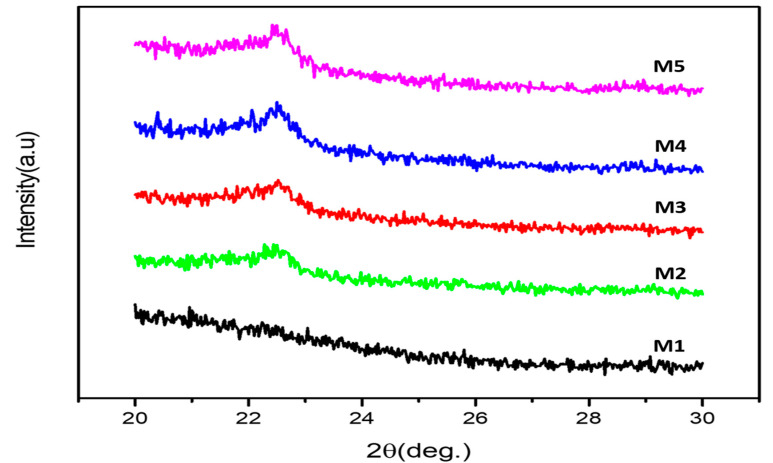
XRD of PLA, PLA-PBS, and PLA-PBS-MCC composite films.

**Figure 6 polymers-13-00465-f006:**
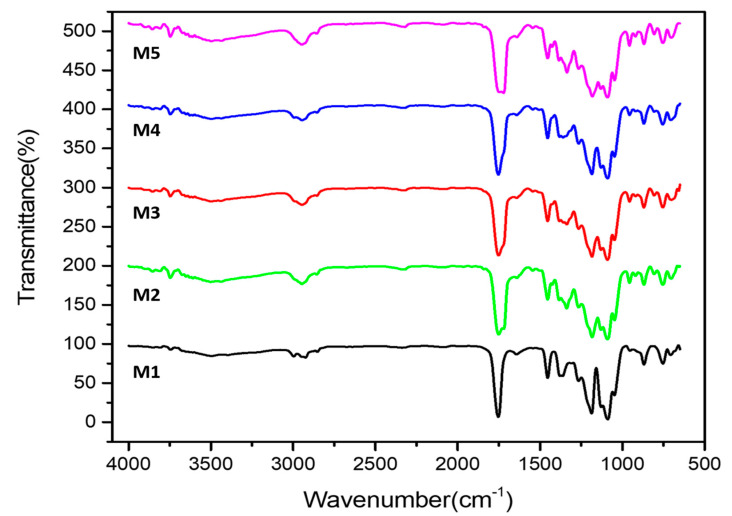
FTIR of PLA, PLA-PBS, and PLA-PBS-MCC composite films.

**Figure 7 polymers-13-00465-f007:**
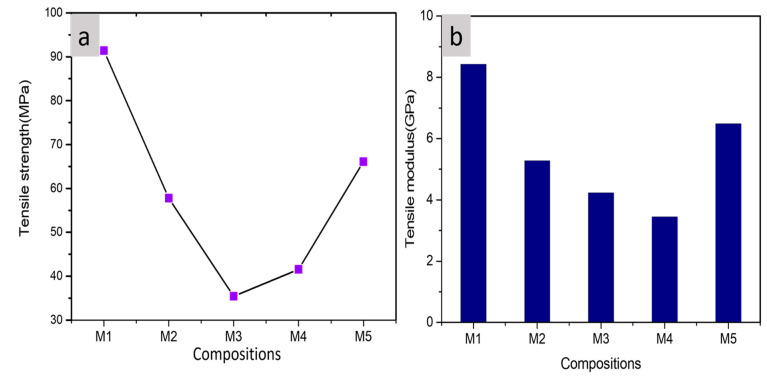
(**a**) Tensile strength (MPa), (**b**) tensile modulus (GPa), of PLA, PLA-PBS, and PLA-PBS-MCC composite films.

**Figure 8 polymers-13-00465-f008:**
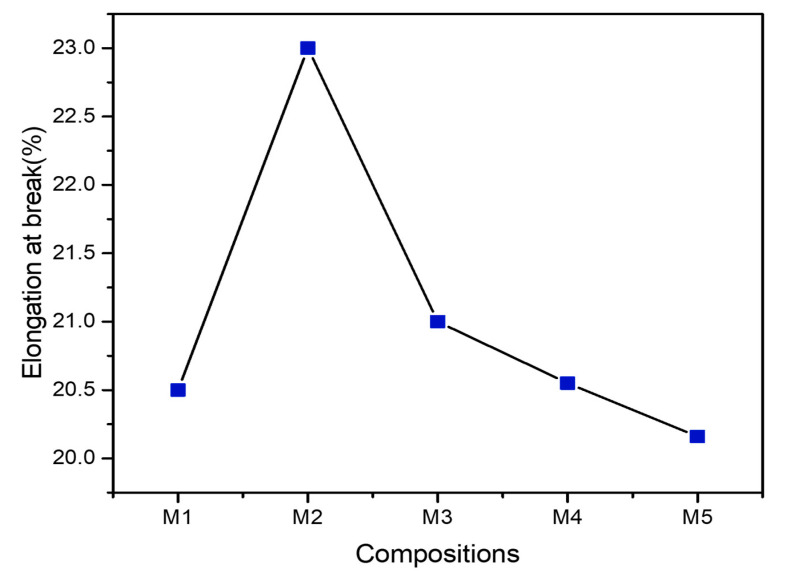
Elongation at break(mm) of PLA, PLA-PBS, and PLA-PBS-MCC composite films

**Figure 9 polymers-13-00465-f009:**
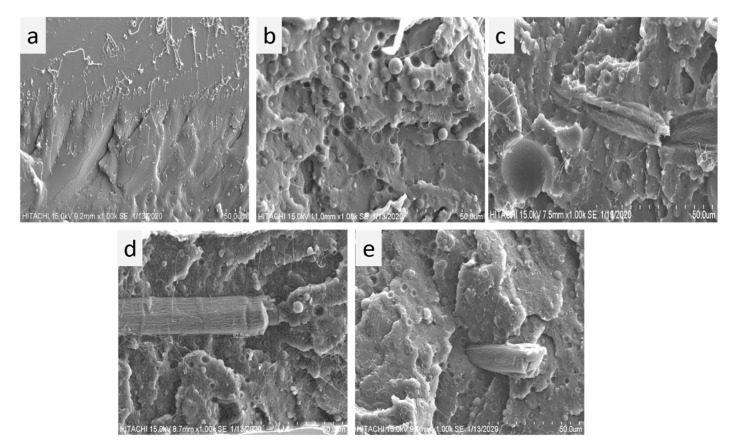
SEM morphology of the fractured surfaces of composite films: (**a**) M1, (**b**) M2, (**c**) M3, (**d**) M4, (**e**) M5.

**Table 1 polymers-13-00465-t001:** Composition of PLA/PBS/MCC Composites

S.No	Composite	PLA (wt%)	PBS (wt%)	MCC
1.	M1	100	0	0
2.	M2	80	20	0
3.	M3	80	20	0.5
4.	M4	80	20	1
5.	M5	80	20	1.5

**Table 2 polymers-13-00465-t002:** Fabrication Process Parameters.

Process	Parameters			
	Temperature (°C)	Pressure (MPa)	Time (min)	Speed (rpm)
Melt mixing	180	-	15	60
Hot pressing	180	150	3	-

**Table 3 polymers-13-00465-t003:** TGA results of PLA/PBS/MCC composite films

Samples	Ti ^a^ (°C)	T50 ^b^% (°C)	Tmax ^c^ (°C)	Tf ^d^ (°C)	Wi ^e^ (°C)	Wmax ^f^ (%)	Wfinal ^g^ (%)	Wresidue ^h^ (%)
M1	284.81	359.42	365.24	386.09	98.9	27.57	1.698	0.372
M2	286.66	361.197	365.72	416.72	98.95	22.76	2.131	1.044
M3	288.12	362.96	365.10	417.74	97.65	22.04	2.221	1.046
M4	289.22	363.11	366.50	419.65	96.43	21.65	2.163	1.064
M5	289.45	362.48	366.72	420.67	95.88	21.21	2.056	1.099

^a^ TGA; initial degradation temperature, ^b^ TGA;50% degradation temperature, ^c^ DTG; peak temperature, ^d^ TGA; final degradation temperature, ^e^ TGA: initial weight loss, ^f^ DTG maximum weight loss, ^g^ TGA; final weight loss, ^h^ TGA char residue weight.

**Table 4 polymers-13-00465-t004:** DSC results of PLA, PLA-PBS, PLA-PBS-MCC composites

Sample	T_g_ (°C)	T_m_ (°C)	ΔH (J/g)	T_m1_ (°C)	ΔH (J/g)	T_m2_ (°C)	ΔH (J/g)
M1	64	-	-	151	2.81	165.14	0.32
M2	62	113.94	6.47	146.21	6.58	152.89	4.28
M3	63	113.60	10.68	146.23	6.74	152.73	5.29
M4	63.10	113.51	10.62	146.33	7.77	152.84	2.41
M5	63.24	113.56	9.34	146.01	8.27	152.68	5.61

## Data Availability

Not applicable.

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
