# Peer review of "Morphology, Structural, Thermal, and Tensile Properties of Bamboo Microcrystalline Cellulose/Poly(Lactic Acid)/Poly(Butylene Succinate) Composites"

_polymers, 2021, doi:10.3390/polym13030465_

Round 1

Reviewer 1 Report

What type of mixer was used for preparation of composites?

Section 2.2.1. has apparently wrong name.

What was the crosshead speed during Young's modulus determination?

How MCC could enhance the interfacial adhesion between PLA and PBS? It should be explained in the paper.

"coke formed by decomposed MCC as earlier revealed to be uniformly distributed to the inside and surface of PLA-PBS-MCC composites, that efficiently impeded the discharge of decomposition products of PLA-PBS blends"

Any photograph of this effect?

How Authors determined the initial degradation temperature?

What mean the initial and final weight loss? Because for sample M1, they are adding to the amount exceeding 100%, same for M2. Seems strange.

Figure 2c indicates that for sample M2 the final residue is lower than for M1, while in Table 3 it is stated otherwise. Requires clarification. 

Figure 3b shows that Tmax is lower for M3 compared to M1, while in Table 3 it is stated otherwise. Requires clarification.

Authors should present the DSC analysis in Table and present the temperatures of transitions, as well as the values of enthalpy, which could be used, e.g. to calculate the degree of crystallinity. 

Authors have to explain the binding ability of MCC in prepared blends.

Manuscript is poorly organized with a lot of linguistic mistakes, which have to be corrected.

Author Response

Reviewer -1

The paper presents an environmentally friendly biodegradable polymer blend with comparable tensile and thermal properties with the synthetic plastics. It is a topic of interest to the researchers in the related areas, but the paper needs some improvement before acceptance for publication. My detailed comments are as follows:

  1. As rightly authors mentioned in the section of Methods, “MCC” were successfully extracted from bamboo fibre by acid hydrolysis technique. Did Authors use the SEM or TEM to prove that it is micron-size cellulose (Microcrystalline cellulose, MCC) rather than ordinary cellulose? In addition, why us “Storage Modulus (E')” as the title for Section 2.2.1.

The micro size of MCC was detected by SEM (Figure 1(f,g,h)) as added in the revised manuscript. The title of section 2.2.1 was a typo-mistake and was corrected as “Extraction of MCC from bamboo fibre”.

  1. As for MCC of higher yield of 80%, how did the authors measure or estimate it? Is it based on total cellulose content of bamboo fibre, or based on the weight of pulp or others?

The total yield was calculated using the total cellulose content from bamboo fibre and it was carried out in our previous published work, the reference#23 of that has been provided in the revised manuscript.

  1. As the author mentioned, MCC could improve the compatibility of PLA with PBS, but how did authors ensure that MCC is evenly dispersed well in the blend within 15 min at 60 rpm? Since dried MCC is hard to re-disperse as far as we know.

As MCC’s extracted were ultrasonicated thus well dispersed and of higher quality already before their addition to the polymer mix. Further, during melt mixing process (180°C,60rpm,15min), mixing and melting of polymers takes place simultaneously ensuring proper mixing of polymers and MCC’s.

  1. The results presented are very detailed from a thermal properties point of view, but in the comments on Table 3, Page 7, authors wrote: “From Table 3, Wmax increases with the addition of PBS then on addition of MCC it reduces insignificantly.” However, I found that Wmax was decreased with the addition of PBS. So, it is not clear.

 As far the statement is concerned, there was a typo mistake otherwise the value of Wmax reduces with the addition of PBS and on addition of MCC it further reduces as clear from DTG(Fig.3) as well as Table 3. Accordingly the statement has been corrected in the revised manuscript.

  1. As for the mechanical properties (Figure 7 and Figure 8), especially elongation at break, presented in Figure 8, Page 11, authors think that the elongation at break increases on addition of PBS and further decreases with addition of MCC insignificantly. But all the data changes only within 2%, it is likely within the experimental error range. How many parallel experiments were tested in the mechanical properties assay? Why was a statistical analysis not performed to ensure the difference between treatments? What is the standard deviation of the tensile properties for each sample?

 We agree with this comment, in our study there was an insignificant effect of PBS and MCC addition on the elongation at break of the composites, if we consider the error range too, they all will have nearly similar values even if tests were repeated several times, but for the sake of comparison it was presented in the form of graph. With Limited data, its difficult to do Statistical analysis and generally in Composites very rarely you find it.

  1. Please ensure that the order of the figures and tables in the manuscript is consistent with the order mentioned in the main text. Such as page 5 “Table 5”, page 11 “Figure 8(a)”, page 12 “Figure 8(a)” and “Figure 8(b-e)” should be typo mistake.

We agree that there were typo mistakes with regards to mentioning of numbering of Tables and figures on page (5,11 &12) and Now we  corrected it.

Reviewer 2 Report

The paper presents an environmentally friendly biodegradable polymer blend with comparable tensile and thermal properties with the synthetic plastics. It is a topic of interest to the researchers in the related areas but the paper needs some improvement before acceptance for publication. My detailed comments are as follows:

  1. As rightly authors mentioned in the section of Methods, “MCC” were successfully extracted from bamboo fibre by acid hydrolysis technique. Did Authors use the SEM or TEM to prove that it is micron-size cellulose (Microcrystalline cellulose, MCC) rather than ordinary cellulose? In addition, why us “Storage Modulus (E')” as the title for Section 2.2.1.

  1. As for MCC of higher yield of 80%, how did the authors measure or estimate it? Is it based on total cellulose content of bamboo fibre, or based on the weight of pulp or others?

  1. As the author mentioned, MCC could improve the compatibility of PLA with PBS, but how did authors ensure that MCC is evenly dispersed well in the blend within 15 min at 60 rpm? Since dried MCC is hard to re-disperse as far as we know.

  1. The results presented are very detailed from a thermal properties point of view, but in the comments on Table 3, Page 7, authors wrote: “From Table 3, Wmax increases with the addition of PBS then on addition of MCC it reduces insignificantly.” However, I found that Wmax was decreased with the addition of PBS. So, it is not clear.

  1. As for the mechanical properties (Figure 7 and Figure 8), especially elongation at break, presented in Figure 8, Page 11, authors think that the elongation at break increases on addition of PBS and further decreases with addition of MCC insignificantly. But all the data changes only within 2%, it is likely within the experimental error range. How many parallel experiments were tested in the mechanical properties assay? Why was a statistical analysis not performed to ensure the difference between treatments? What is the standard deviation of the tensile properties for each sample?

  1. Please ensure that the order of the figures and tables in the manuscript is consistent with the order mentioned in the main text. Such as page 5 “Table 5”, page 11 “Figure 8(a)”, page 12 “Figure 8(a)” and “Figure 8(b-e)” should be typo mistake.

Author Response

Reviewer -2

Comments and Suggestions for Authors

  1. What type of mixer was used for preparation of composites?

            High Speed Brabender mixer

  1. Section 2.2.1. has apparently wrong name.

We are sorry for the typo-mistake, it has been corrected in the revised manuscript.

  1. What was the crosshead speed during Young's modulus determination?

The crosshead speed during tensile testing was 10 mm/min

  1. How MCC could enhance the interfacial adhesion between PLA and PBS? It should be explained in the paper.

It has been explained in the section 3.6 that the improvement in the cohesive forces at the interface of MCC and the PLA matrix occurs as a result of MCC addition; thus, matrix interface became more miscible as a result increases the compatibility of PLA with PBS.

  1. "coke formed by decomposed MCC as earlier revealed to be uniformly distributed to the inside and surface of PLA-PBS-MCC composites, that efficiently impeded the discharge of decomposition products of PLA-PBS blends"

Any photograph of this effect?

Although we have no photographic proof of this process but it has been observed by several researchers as added in the revised manuscript in section 3.2 as” Similar effect was observed for PLA-PBS-MCC blend [29] and polypropylene (PP)/MCC composites, where MCC effectively hindered the release of decomposition products of the polymers [17,30]”

  1. How Authors determined the initial degradation temperature?

The initial degradation temperature is the point in the TGA graph where it starts declining and it was obtained from TGA data as well as from Figure 2(b), showing the zoomed part of initial degradation of all composites.

  1. What mean the initial and final weight loss? Because for sample M1, they are adding to the amount exceeding 100%, same for M2. Seems strange.

The term “initial and final weight loss” is not appropriate instead it should be “weigh loss at initial and final degradation temperature”. The initial weight loss corresponds to weight loss at initial degradation temperature and final weight loss correspond to final degradation temperature.

  1. Figure 2c indicates that for sample M2 the final residue is lower than for M1, while in Table 3 it is stated otherwise. Requires clarification. 

The results have been synchronized in the revised manuscript. Indeed M2 is less than M1,there was an error that was removed and corrected.

  1. Figure 3b shows that Tmax is lower for M3 compared to M1, while in Table 3 it is stated otherwise. Requires clarification.

I agree with the reviewers comment that Tmax is lower for M3 compared to M1, thus it has been corrected in accordance with Figure 3b.

  1. Authors should present the DSC analysis in Table and present the temperatures of transitions, as well as the values of enthalpy, which could be used, e.g., to calculate the degree of crystallinity. 

       Table for DSC analysis has been added in the revised manuscript as “Table 4”.

  1. Authors have to explain the binding ability of MCC in prepared blends.

The binding ability of MCC was briefly explained in section 3.5 as, “The binder functionality of MCC depends upon its ability to deform plastically on application of compressive force [49]. MCC particles form hydrogen bonds leading to strong compacts and their critical properties corresponding to their functionality as a binder include particle size, moisture content, bulk density, specific surface area, and crystallinity [49]”. Also, it was further explained in section 3.6.

  1. Manuscript is poorly organized with a lot of linguistic mistakes, which have to be corrected.

The revised manuscript has been organized well and all the necessary typo and grammatical mistakes were corrected.

Round 2

Reviewer 1 Report

I see that Authors adressed all of my comments properly, therefore from my side the paper could be accepted.

Reviewer 2 Report

The paper has been revised and the authors have responded to my question.